# Model Test Study of the Synergistic Interaction between New and Existing Components of Sheet Pile Walls

Wenhui Zhao, Xiaomin Wu * and Xuening Ma

School of Civil Engineering, Lanzhou Jiaotong University, Lanzhou 730070, China
* Correspondence: wuxiaomin1226@163.com

**Abstract:** New and existing components of retaining structures are often combined in the width section. When combining the design and use requirements of the existing and new structures, the synergistic interactions between the existing and new structures and the design and working conditions require clarification. In conjunction with an actual project, a sheet pile wall consisting of existing and new components is proposed to retain an embankment. Indoor model tests were carried out to simulate the excavation and compaction and investigate changes in earth pressure, pile bending moment, shear force, and load-sharing ratio of the new and existing sheet pile walls at different stages. The results show that the earth pressure of the cantilever section of the existing and new piles increases with an increase in the fill volume or the upper uniform load. An inflection point is observed in the earth pressure curve halfway between the pile top and the ground due to sudden changes in the pile and soil stiffness. The bending moment of the new and existing piles increases and decreases with the distance from the top of the pile under different working conditions, and the maximum bending moment occurred at 0.485 and 0.9 m from the bottom of the existing pile and the bottom of the new pile, respectively. The lateral displacement of the new and existing piles decreases with the distance from the top of the pile. Due to the adjustment of the structural force in the cantilever section and the soil reaction force in front of the pile, the displacement curves of the new and existing piles are similar in the cantilever section. The displacement in the anchored section is initially larger for the existing pile than for the new pile but then becomes similar for both piles. In working condition 5, the top displacement of the existing pile was 6.531 mm, exceeding the control value (5.6 mm). The earth-pressure-sharing ratio of the existing pile decreases with an increase in the width of the filling material or the load. When the load was applied, the earth-pressure-sharing ratio of the existing pile was 0.451, indicating that the structural design of the combined sheet pile wall is reasonable.

**Keywords:** earth pressure; load-sharing ratio; model test; sheet pile walls; synergistic interaction

## 1. Introduction

For improvements to China's railway network, new railway sections are connected to existing railway sections. Therefore, construction near existing lines is increasing every year [1,2]. When a subgrade structure is widened, geosynthetic materials, lightweight fillers, and new retaining structures are required to minimize the impact of the new embankment structures on existing structures [3,4]. The integration of the new and existing sheet-pile-wall retaining structures is challenging. The new reinforcement structure cannot affect road operations. The new and existing structures should have high structural strength to meet the force and deformation requirements. It is critical to clarify the force mechanisms of the new and existing retaining structures and the synergistic interactions.

The sheet pile wall is an improvement on the anti-slip pile to retain a slope. Retaining plates are placed between the anti-slip piles. The sheet pile wall retains an embankment by transferring the earth pressure of the cantilevered section to the foundation anchored in the ground. The lateral displacement of the pile causes the resistance of the foundation soil in

front of the pile to achieve a mechanical balance [5]. Tang [6] analyzed the influences of the pile length, stiffness, position, and soil parameters on the synergistic effect of a sheet pile wall using numerical simulations and field measurements. An analysis of the earth pressure on both sides of the pile body indicated that the optimal pile length was 14 m. Zhang [7] analyzed the influences of the pile stiffness and the anisotropy of soft clay on the lateral earth pressure of the sheet pile wall using numerical calculations. In a cantilever sheet pile wall, the displacement of the pile top commonly exceeded the standards, requiring a reduction in the size of the pile section. Fall [8] analyzed the changes in the pile bending moment, pile displacement, and bolt reaction force of a double-anchor sheet pile wall during the excavation of the surrounding soil and proposed optimization parameters. Xu [9] used numerical analysis to investigate the horizontal displacement and bending moment of sheet pile walls with different pile lengths, optimized the number and length of short piles, and designed a reliable and cost-effective retaining structure. Ohori [10] and Sawaguchi [11] each proposed a calculation model for a double-row sheet pile wall under horizontal loads and used model tests to obtain the wall deformation and bending moments to verify the reliability of the calculation models. Combined with an actual project, Finno [12] analyzed the function of a stepped pile group and optimized the parameters. Kannaujiya [13] used numerical modelling to analyze the mechanical behavior of an anchored sheet pile wall during excavation. Different types of sheet pile walls and structural combinations are used in different terrains. This study proposes a sheet pile wall consisting of new and existing components for the third and fourth lines of the Zhongwei–Lanzhou Passenger Line connected to the Lanzhou–Xinjiang Line, as shown in Figure 1. The purpose of the project is to ensure that the superstructure has sufficient width. Differently from finite element and discrete element analysis methods, laboratory model testing has guiding significance [14,15].

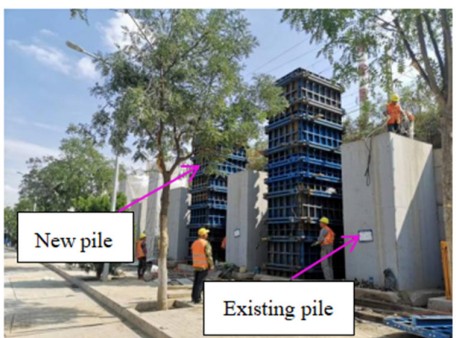 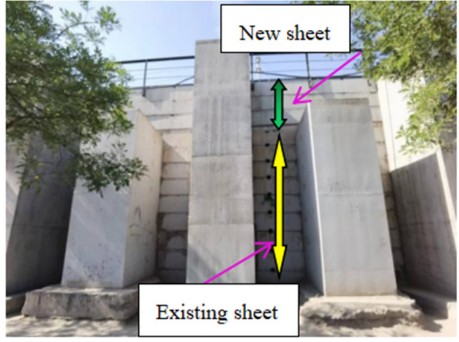

**Figure 1.** The sheet pile wall consisting of new and existing components.

A laboratory model test was carried out to simulate the excavation and compaction. The earth pressure distribution of the composite retaining structure was investigated in different construction stages, and the internal force and horizontal displacement of the composite sheet pile wall were analyzed. The earth-pressure ratio between the existing pile and the new pile was determined. The synergistic interaction between the new and existing components was evaluated. The research results can provide technical support for the construction of the new and existing components of sheet pile walls used in railway-engineering applications.

## 2. Similarity Theory and the Model's Test Parameters

### 2.1. The Model's Test Parameters

This study investigates the new and existing sheet pile walls of the Zhongwei–Lanzhou Railway in the DK7 + 831.755~DK9 + 703.0 section. The subgrade cross-section with a long cantilever section is shown in Figure 2. As shown in Figure 1, the new pile is made of C40 reinforced concrete, with dimensions of 1.5 m × 2.0 m × 24 m. The pile spacing is 6 m. The new sheet consists of precast C40 reinforced concrete with a height of 0.5 m and

a thickness of 0.4 m. The existing pile consists of C40 reinforced concrete with dimensions of 1.5 m × 2.0 m × 15 m. The pile spacing is 6 m.

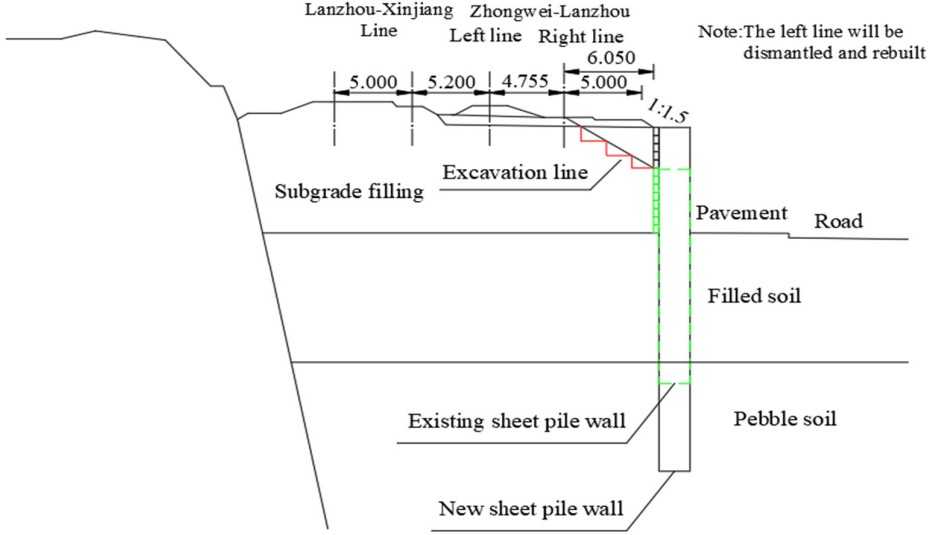

**Figure 2.** Subgrade cross-section (unit: m).

Based on similarity theory [16], the similarity ratio of the model, $C_l$, is 1/8; the mass density ($C_\rho$) is 1; the modulus of elasticity, $C_E$, is 1; the Poisson's ratio, $C_\mu$, is 1; the reinforcement ratio, $C_p$, is 1; the material capacity, $C_\gamma$, is 1; the line load, $C_q$, is 1/8; the stress, $C_\sigma$, is 1; the concentrated load, $C_F$, is 1/64; the displacement, $C_S$, is 1; and the strain, $C_\varepsilon$, is 1.

The cross-sectional size of the middle pile is 0.19 m × 0.25 m, and that of the side pile is 0.10 m × 0.25 m. The length of the existing pile is 1.875 m, and that of the new pile is 3.0 m. The spacing of the model piles is 0.75 m. The height of the model sheet is 0.06 m, and its thickness is 0.05 m. The concrete strength grade of the pile body is C40. Each pile is reinforced with seven φ10 mm reinforcement bars, including 5 on the back side and 2 on the front side, to achieve an equal reinforcement ratio.

The stress influence zone at the bottom of the pile is 2–3 times the pile diameter [17,18]. The subsoil thickness of the model pile is 0.6 m to minimize the influence on the constraint at the bottom of the pile. The model has three existing piles and two new piles in the middle to reduce the influence of the boundary constraint on the pile (see in Figure 3).

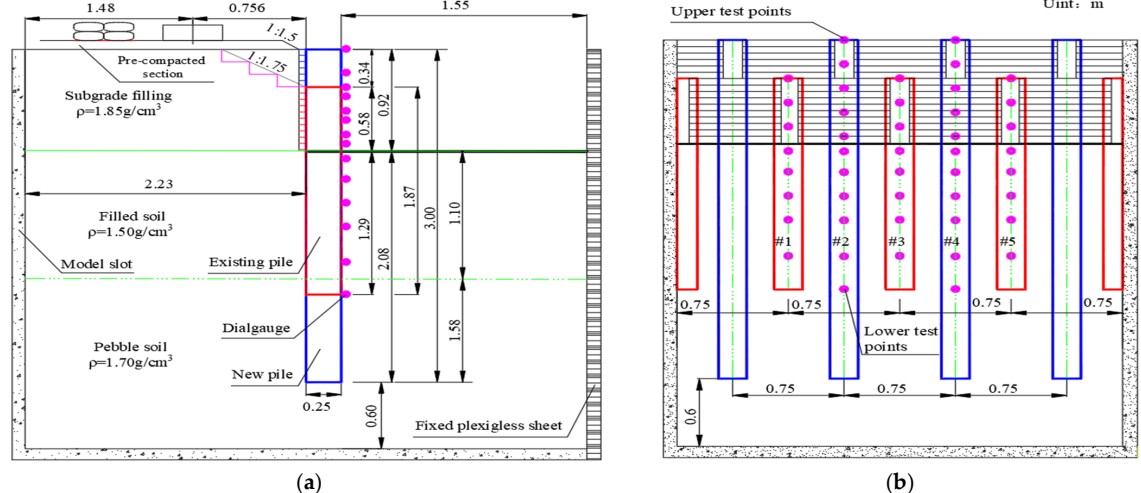

**Figure 3.** Diagram of the model (unit: m). (**a**) Cross-sectional view. (**b**) Side view.

### 2.2. Test Component Configuration

(1) Configuration of pile strain gauges

The strain value of the pile was measured to determine the stress of the pile, and the bending moment during filling and loading was calculated. The three existing piles in the middle (Pile #1, Pile #3, and Pile #5) and the two new piles in the middle (Pile #2 and Pile #4) were tested to prevent a boundary effect. The configuration of the pile strain gauges is shown in Figure 4.

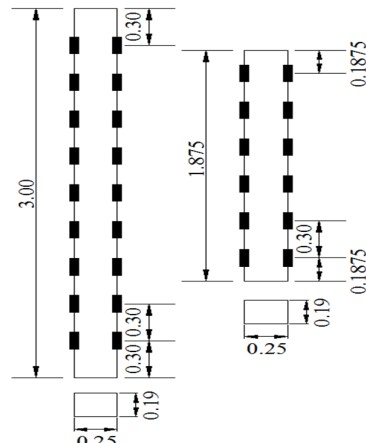

**Figure 4.** Configuration of pile strain gauges (unit: m).

(2) Pressure-sensor configuration

Load sensors (Kingmach Measurement And Monitoring Technology Co., Ltd., Changsha, China) were installed between the sheet and the pile to test the horizontal force transmitted by the retaining sheet to the pile and determine the earth pressure behind the sheet. Measurements were obtained during the filling and loading from sensors installed on the sheets between the middle piles. The load sensor configuration behind the sheet is shown in Figure 5.

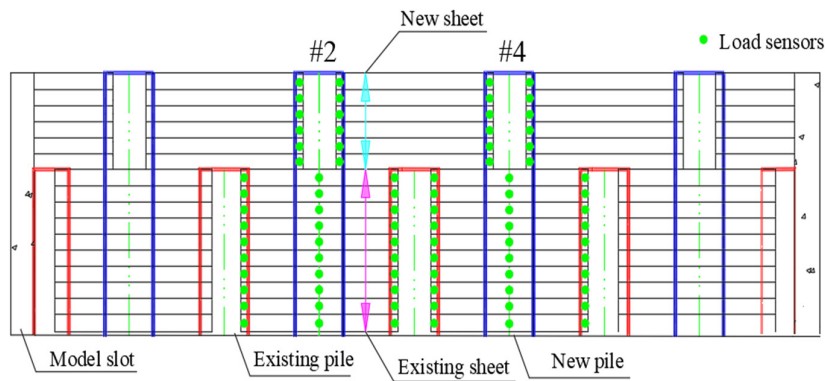

**Figure 5.** Load-sensor configuration behind the sheets.

(3) Horizontal displacement test of the pile

The displacement of the pile was obtained in each loading stage from displacement meters (Kingmach Measurement And Monitoring Technology Co., Ltd., Changsha, China) in the cantilever section of the pile and in the filling soil. The displacement meters were mechanical dial gauges (accuracy of 0.01 mm) each attached to the frame beam with a magnetic bracket.

### 2.3. Geotechnical Testing

Sieving tests, compaction tests, and large shear tests of the filling material were carried out. The results are listed in Table 1. The large shear test equipment is shown in Figure 6. The shear test was conducted with different normal stresses at a shear rate of 0.25 mm/min. The test was stopped when the peak shear strain reached 6%.

**Table 1.** The parameters of the filling material.

| Type | Grading Test | | Compaction Test | | Shear Test | |
|---|---|---|---|---|---|---|
| | $C_u$ | $C_c$ | $W_o$/% | $\rho_o$/(g/cm$^3$) | c/kPa | $\varphi$/° |
| Subgrade filling | 49 | 0.3 | 5.7 | 2.05 | 13.82 | 35.4 |
| Filling soil | — | — | 16.4 | 1.81 | 10.1 | 35.3 |
| Pebble soil | 23.5 | 1.15 | 8.5 | 2.03 | 5.2 | 36.1 |

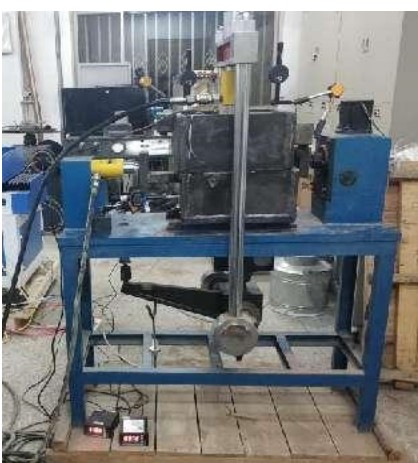

**Figure 6.** The large shear test's equipment.

### 2.4. Model Filling and Compaction

The filling and compaction during the model were conducted as follows.

(1) The bottom layer was filled with pebble soil using controlled compaction of 0.85 (to achieve a density of 1.7 g/cm$^3$). Filling occurred in layers to a height of 0.6 m.
(2) The new and existing piles were placed in their locations.
(3) The pebble soil was filled to a height of 1.58 m and compacted to the specified density.
(4) The filling soil was added and compacted in three layers to a thickness of 1.1 m and a density of about 1.5 g/cm$^3$ (compaction of 0.85). The moisture content was 12%.
(5) The subgrade fill was compacted in layers, and the sheets (existing sheets) were installed at the height of the existing piles.
(6) The subgrade filling was completed to obtain a slope ratio of 1:1.75.
(7) The stacked freight line load was pre-pressed to consolidate the soil. The remaining new piles were installed and tested until their strength satisfied the design requirements.
(8) The subgrade filling soil was added to achieve the desired width, and the retaining plates of the new pile were added. The loading devices were installed.

### 2.5. Loading and Data Collection

According to *the Code for the Design of the Subgrade of Railways* (TB 10001-2016), the track is a high-speed railway ballast track, and the load on the top surface of the subgrade should meet those requirements. According to *the Code for the Design of Retaining Structures of Railway Subgrades* (TB 10025-2019), the displacement of the pile's top should be less than 1/100 of the length of the pile cantilever and should not exceed 10 cm. The cantilever end

lengths of the existing and new piles in this test were 0.56 and 0.92 m, respectively. Thus, the displacements were to be less than 5.6 and 9.2 mm, respectively.

(1)    Model test loading scheme

The ZK load of the new line was 36.8 kPa, the self-weight load on the track was 17.3 kPa, the total load was 54.1 kPa (P), and the width was 0.425 m. When only the existing sheet pile wall was loaded, the freight line load was preloaded with 68.4 kN/m². The freight line load was removed and the soil was backfilled after the new piles were installed. The line was then loaded in two steps to a design load of 54.1 kN/m². In accordance with the conversion relationship, the top load was applied to the subgrade at 1.5 P→2.0 P→2.5 P→ until the pile top displacement was exceeded (with an increment of 0.5 P). The pile deformation, load, and pile strain were measured at 5, 15, 30, 45, and 60 min after each load level was applied, and subsequently at 30 min intervals. The deformation of the pile top did not exceed 0.1 mm per hour, and the pile deformation could be considered stable when the above-mentioned deformation occurred twice in succession. When the deformation rate of the pile top stabilized, the next load level was applied, as shown in Figure 7.

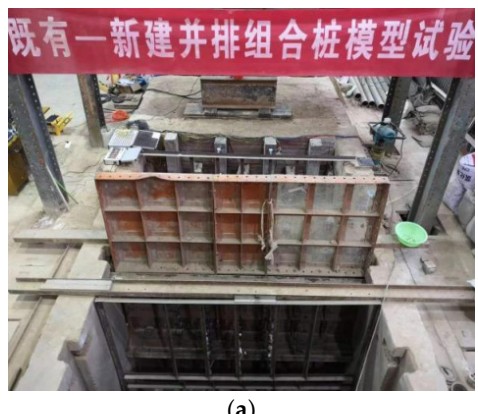
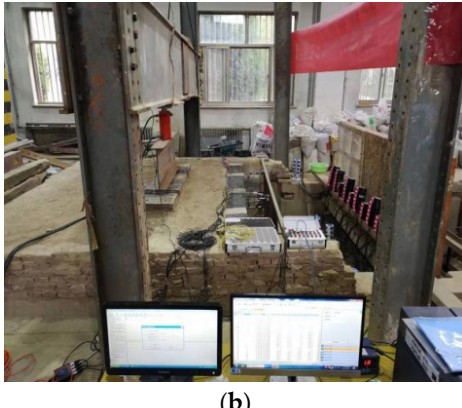

| (**a**) | (**b**) |
|---|---|

**Figure 7.** The model test: (**a**) horizontal view (**b**) side view.

(2)    Test data collection and processing

The strain values were obtained from the static strain test and analysis system. The internal force and the earth pressure behind the composite sheet pile wall were calculated as follows.

①    Calculation of pile bending moment

The pile curvature *φ(z)* at relatively small strains can be obtained using the elastic foundation beam theory [19]:

$$\varphi(z) = \frac{\Delta\varepsilon}{a} \tag{1}$$

where *z* is the distance from the pile top to the strain gauge (unit: m). $\Delta\varepsilon$ is the strain difference between the back of the pile and the surface of the pile at depth *z* derived from the strain gauges. *a* is the height of the pile section, which was 0.25 m for the test pile.

The relationship between the pile curvature and the bending moment gives the bending moment of the model pile at different sections:

$$M(z) = EI \cdot \varphi(z) \tag{2}$$

where *EI* is the flexural stiffness of the test pile, kN·m².

② Earth pressure calculation

The force *F* acting on the pile is obtained from the load sensor installed between the retaining plate and the pile. If the earth pressure on each layer is uniform, it can be converted to *p* [7]:

$$p = \frac{2F}{L \cdot a} \tag{3}$$

where *L* is the span of the retaining plate (unit: m). *a* is the width of the retaining plate (unit: m).

## 3. Results and Discussion

After the completion of each test phase of the model test, including component installation, model filling, and the completion of composite retaining structure, a loading simulation was carried out, and five working conditions were evaluated.

Working condition 2: Subgrade width fill load.
Working condition 3: New line superstructure and train load (P).
Working condition 4: 2 P load.
Working condition 5: 4 P load (the pile displacement was exceeded when the upper evenly distributed 4 P load was applied).

Only piles #2 (new pile) and #3 (existing pile) were selected for this analysis.

### 3.1. Earth Pressure Analysis of the Composite Structure

Figure 8 shows the earth pressure on the cantilevered sections of the existing and new piles under different working conditions. For working condition 1, the earth pressure of the existing pile increased and decreased with the distance from the top of the existing pile. The maximum earth pressure was 3.019 kPa at about one third from the ground line. The earth pressure was lower than the passive earth pressure and higher than the static earth pressure at each position. Field measurements have shown a similar trend for the earth pressure [20]. The main reason is that the soil behind the wall undergoes stress during mechanical rolling or vibration, resulting in earth pressure increases. Under working condition 2, the earth pressure of the existing pile became higher as the upper backfill load increased. The earth pressure decreased as the distance from the top of the existing pile increased. It was equal to the static earth pressure at the top and the active earth pressure at the bottom. However, the earth pressure of the new pile increased with the distance from the top. The earth pressure from the new pile top to the existing pile top was slightly higher than the static earth pressure. In contrast, the earth pressure decreased from the top of the existing pile to the ground line and was lower than that of the existing pile. The maximum earth pressure occurred near the top of the existing pile (1.797 kPa). Under working condition 3, the earth pressure of the existing pile increased with the distance from the top of the pile under a uniform load. It affected the lower part of the cantilever, and the maximum earth pressure was 7.367 kPa. In contrast, the earth pressure from the top of the new pile to the top of the existing pile was small. The new piles had full contact with the existing sheets from the top of the pile to the ground. Since the new piles and the existing piles have similar lateral bearing capacities, the maximum earth pressure occurs at the same position on the new and existing piles. The maximum earth pressure was 6.540 kPa. The earth pressure was higher in working conditions 4 and 5 than in the other working conditions at all locations on the existing pile. The earth pressure of the new pile increased and decreased as the distance from the new pile top to the existing pile top and from the existing pile top to the ground increased, respectively. The main reason is a sudden change in the composite stiffness of the pile and the soil at the top of the existing pile and the ground. The upper composite stiffness is lower than the lower composite stiffness [21,22]. In working conditions 4 and 5, cracks appeared in the new sheet in the middle and propagated upward (Figure 9a). Figure 9b shows that the vertical and lateral deformation are larger under a uniform upper load. The main reasons are that the soil near the pile cannot be thoroughly compacted due to the narrow space, and the viscoplastic deformation is large due to step excavation

and backfilling. Thus, the retaining structure is adjacent to the void, and the stiffness is low. Transverse deformation and shear cracks caused by the upper load occurred, which is consistent with field-measured data [20,23].

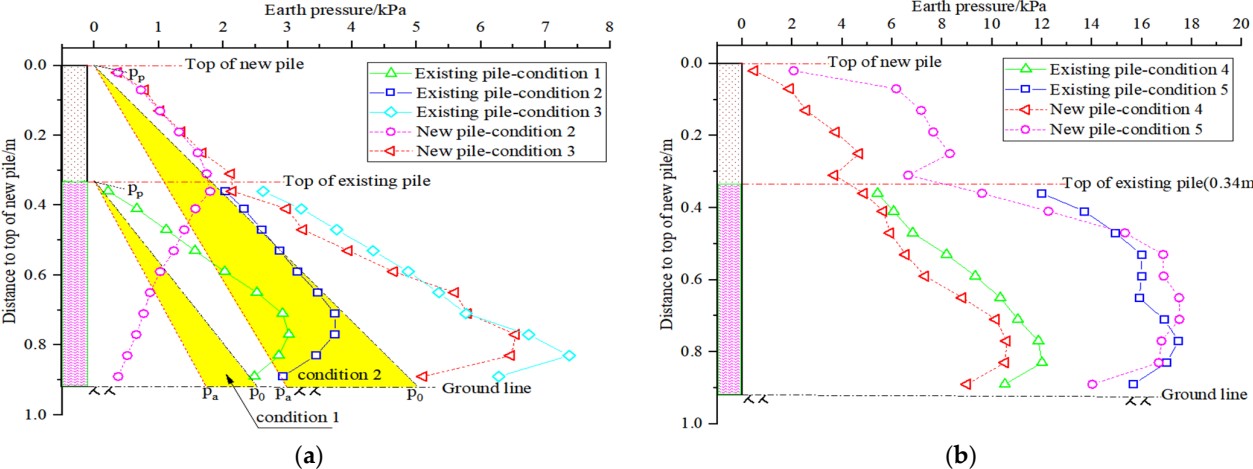

**Figure 8.** Earth pressure on the cantilever sections of the existing and new piles under different working conditions: (**a**) working conditions 1, 2 and 3; (**b**) working conditions 4 and 5.

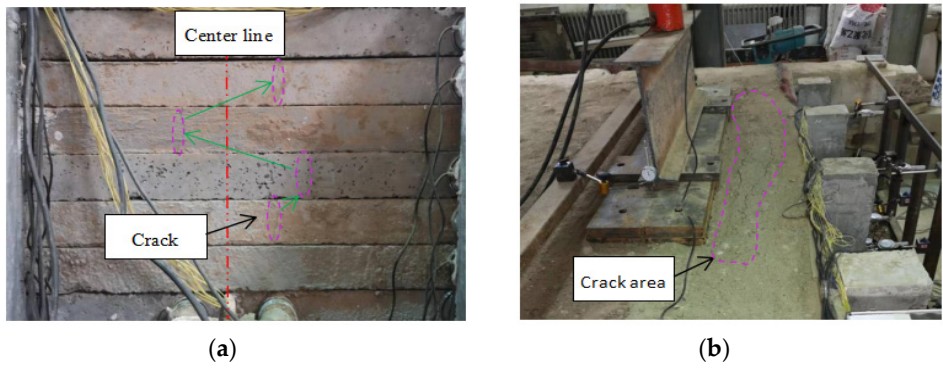

**Figure 9.** Crack development during loading: (**a**) cracks in sheets; (**b**) surface cracks.

Figure 10 shows the additional earth pressure under the strip load. At an upper load amplitude of q, the additional earth pressure of the pile top was slightly different for the new and existing piles. A large difference in the earth pressure was observed between the top of the existing pile and the ground line, and the rate of increase was higher for the new than for the existing pile. At an upper load amplitude of 2q, the rate of increase in the additional earth pressure was similar for the new and existing piles. The new pile had a larger rate of increase than the existing pile at the ground. At an upper load amplitude of 4q, the rate of increase in the additional stress was higher for the new pile than for the existing pile because the anchorage section of the new pile was longer and its resistance capacity was larger. Under a strip load, the additional earth pressure amplitude derived from the 45° distribution approach is typically closer to the measured value than the Beton–Kalender approach [24,25].

### 3.2. Bending Moment of the Piles

Figure 11 shows the bending moments of the existing and new piles under different working conditions. For the existing pile, the bending moment increased and then decreased with the distance in all working conditions. The maximum bending moment of the existing pile was located 0.485 m from the pile bottom, and the values were 2.569, 4.871, 6.156, 12.125, and 23.492 kN·m for the five working conditions, respectively. For the new pile, the bending moment increased and then decreased with distance under working

conditions 2–5. The maximum bending moment was located 0.9 m from the bottom of the pile, and the values were 6.410, 16.831, 31.580, and 68.897 kN·m for working conditions 2–5, respectively. Since the lengths of the existing pile and the cantilever section were short, the bending moment of the existing pile was smaller than that of the new pile under the same working condition and the same position for working conditions 2 to 5.

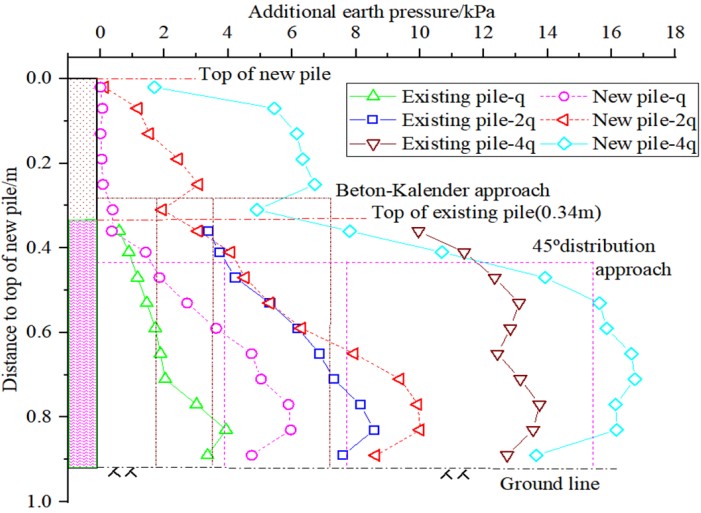

**Figure 10.** Additional earth pressure under a strip load.

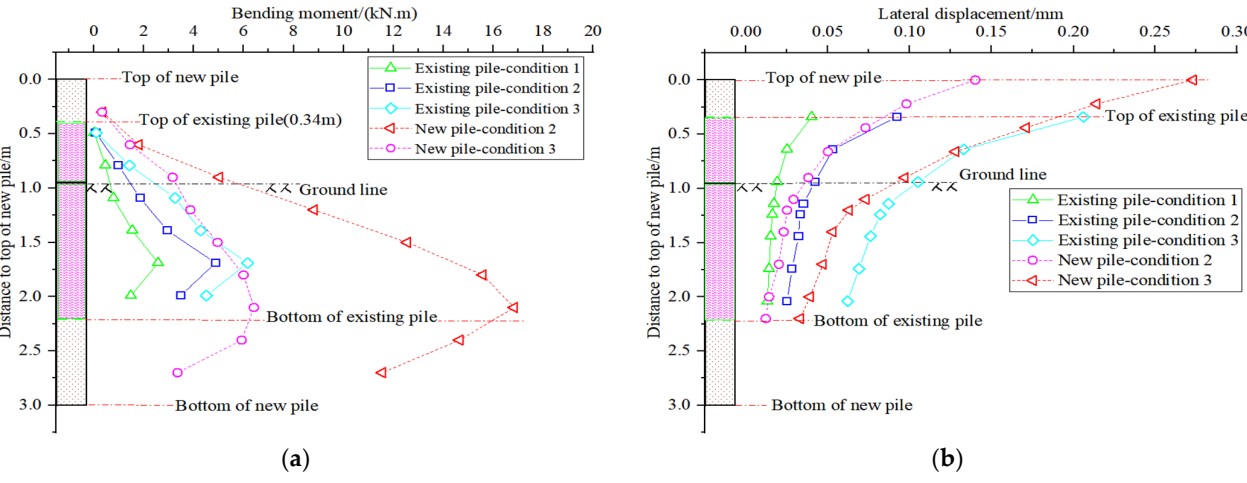

**Figure 11.** Bending moment of existing and new piles under different working conditions: (**a**) working conditions 1, 2, and 3; (**b**) working conditions 4 and 5.

### 3.3. Lateral Displacement of the Piles

Figure 12 shows the displacement of the existing and new piles under different working conditions. The lateral displacements of the new and existing piles decreased with the distance from the pile top for working conditions 1 to 5. The change rate of the lateral displacement was significantly higher in the cantilever section than in the anchorage section. The lateral displacement at the top of the existing pile was 0.04, 0.092, 0.206, 0.684, and 6.531 mm (>control value of 5.6 mm) for the five working conditions, respectively. The lateral displacement at the top of the new pile was 0.14, 0.273, 0.984, and 8.713 mm for working conditions 2 to 5, respectively. The displacement curves of the new and existing piles are similar in the cantilever section in working conditions 2 to 5. The effect of the fill soil and the upper preload on the existing sheet pile wall was small, and the lateral deformation caused by the upper uniform load was large under condition 1. In working conditions 2 and 3, the lateral displacement of the anchorage section was larger for the

existing pile than for the new pile. In working conditions 4 and 5, the lateral displacements of the anchorage sections of the new and existing piles were similar at the same buried depth, indicating high stiffness of the pile. Due to the earth pressure of the cantilever wall, the retaining plate of the existing wall bears more pressure than that of the new wall. The lateral displacement was small in working conditions 1 to 4. When the upper uniform load was applied under working condition 5, the lateral displacements of the existing and new piles increased abruptly, and the displacement of the anchorage section changed significantly, indicating that the resistance of the soil in front of the pile stabilized [26,27].

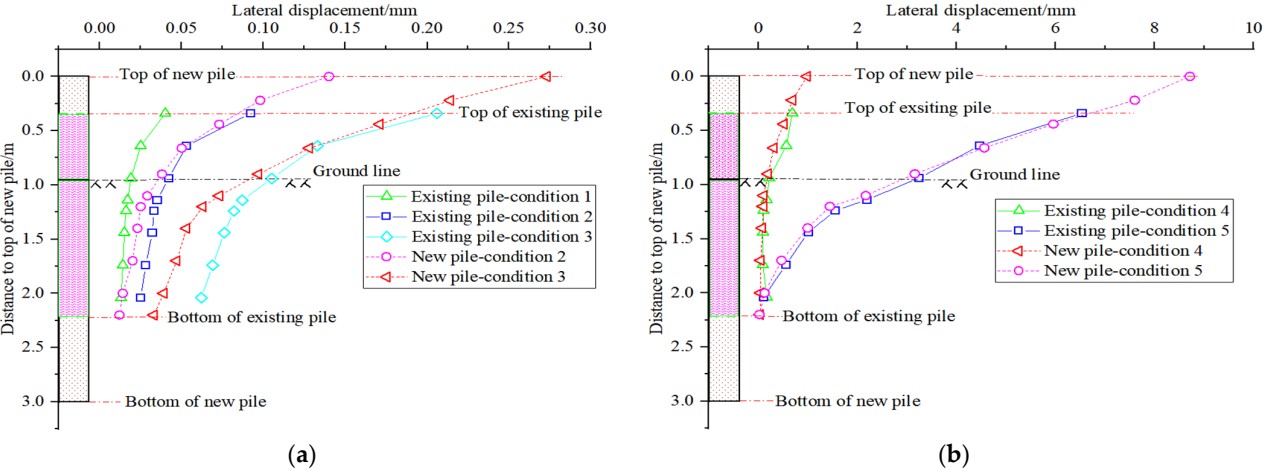

**Figure 12.** Lateral displacement of the existing and new piles under different working conditions: (**a**) working conditions 1, 2, and 3; (**b**) working conditions 4 and 5.

*3.4. Synergetic Mechanism of Combined Structure Analysis*

Before the construction of the new structure, the entire load was borne by the existing structure. After the construction of the new structure, part of the load was transferred to the new structure. Thus, the combined structure shared the load. In the model experiment, $\lambda_1$ and $\lambda_2$ represent the earth-pressure-sharing ratios of the existing and new piles of the combined structure, respectively. They are calculated as follows:

$$\lambda_1 = \frac{P_1}{P_1 + P_2} \tag{4}$$

$$\lambda_2 = \frac{P_2}{P_1 + P_2} \tag{5}$$

where $P_1$ is the earth pressure on the existing sheet pile wall and $P_2$ is the earth pressure on the new sheet pile wall.

Figure 13 shows $\lambda_1$ in the combined structure under different working conditions. The earth-pressure-sharing ratio decreases as the width of the filling material or the uniform upper load increases. In working condition 1, the earth pressure in the cantilever section was borne entirely by the existing structure. After the new structure was added, the earth-pressure-sharing ratio as 0.564. The earth-pressure-sharing ratios of the existing pile were 0.451, 0.446, and 0.403 in working conditions 3 to 5, respectively. These results indicate that the earth pressure changes in the cantilever section of the existing wall when a uniform load is applied. The earth pressure is similar for the new and existing piles at the same height, but the new pile has a higher earth-pressure-sharing ratio. Therefore, it is reasonable to increase the depth of the anchorage section of the new pile so that it can resist lateral deformation if this section has the same height as that of the existing pile.

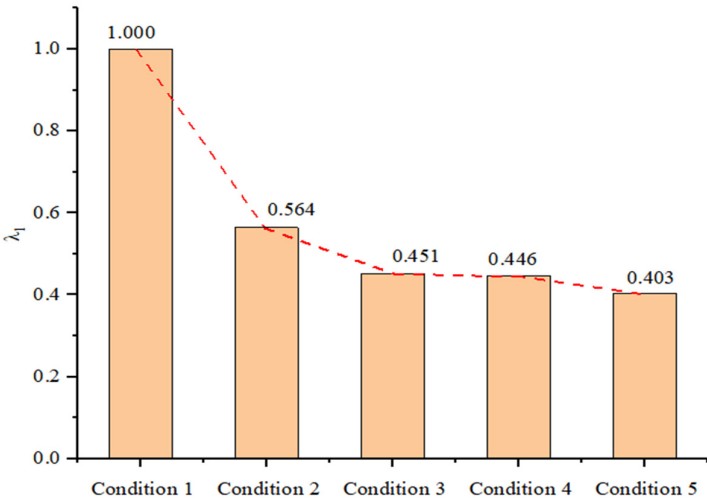

**Figure 13.** $\lambda_1$ in the combined structure under different working conditions.

## 4. Recommendations and Conclusions

### 4.1. Recommendations

The following recommendations can be made for the study of the synergistic interactions between new and existing components of sheet pile walls.

(1) A static test and a dynamic test should be carried out in the field to further study the synergistic interaction of the combined structure.

(2) New technologies and new methods should be used for the combined structure, and the parameters should be optimized [28–30].

### 4.2. Conclusions

Indoor model tests were conducted to simulate the responses of new and existing sections of a combined sheet pile wall. The earth pressure, pile bending moment, shear force, and load-sharing ratio were evaluated under different working conditions. The following conclusions can be drawn.

(1) The earth pressure of the existing and new piles increased with the width of the filling soil or the uniform load on the upper part. The earth pressure of the existing pile increased and then decreased with the distance under all working conditions. In contrast, the earth pressure curve of the new pile had an inflection point halfway between the pile top and the ground.

(2) The bending moments of the new and existing piles increased and then decreased with the distance from the top of the pile under all working conditions. The maximum bending moment occurred at 0.485 and 0.9 m from the bottom of the existing pile and the bottom of the new pile, respectively. The bending moment of the existing pile was smaller than that of the new pile in the same working condition and at the same height.

(3) The lateral displacements of the new and existing piles decreased with the distance from the top of the pile under all working conditions. Its rate of change was significantly higher in the cantilever section than in the anchored section. The lateral displacement of the top of the existing pile was 6.531 mm (>control value of 5.6 mm) in working condition 5, whereas that of the new pile was 8.713 mm (<control value of 9.2 mm).

(4) The earth-pressure-sharing ratio of the existing pile decreased with an increase in the width of the filling soil or the upper load. When an equivalent uniform load of P to 4P was applied, the earth-pressure-sharing ratio of the existing pile ranged from 0.451 to 0.403.

**Author Contributions:** Conceptualization, W.Z. and X.M.; methodology, W.Z. and X.M.; software, W.Z.; formal analysis, W.Z.; resources, X.M.; data curation, X.M.; writing—original draft preparation, W.Z.; writing—review and editing, W.Z. and X.W.; supervision, W.Z. and X.W.; project administration, W.Z.; funding acquisition, W.Z. All authors have read and agreed to the published version of the manuscript.

**Funding:** This work was supported by the Gansu Youth Science and Technology Fund project (grant numbers 21JR1RA249); the Special Funds for Guiding Local Scientific and Technological Development by The Central Government (grant numbers 22ZY1QA005); the Lanzhou Jiaotong University youth science foundation project (grant numbers 2019026). There was no conflict of interest regarding the publication of this paper.

**Institutional Review Board Statement:** This study did not involve humans or animals.

**Informed Consent Statement:** This study did not involve humans.

**Data Availability Statement:** This study did not report any data.

**Conflicts of Interest:** The authors declare no conflict of interest.

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
