# Peer review of "Model Test Study of the Synergistic Interaction between New and Existing Components of Sheet Pile Walls"

_applsci, doi:10.3390/app13031557_

Round 1
Reviewer 1 Report
Following are major revisions
1.Similarity index to be reduced to less than 8% by reviewing manuscript.
2. Keywords to be increased to 5 to 8 and arranged in alphabetical oder
3. Objectives of research study to be clearly stated based on conclusion.
4. Figure 1 to be connected with any other figure where dimensions are given.
5. Following papers to be cited
# Performance of sand granulated rubber mixture for soil stabilization using Discrete Element Method (DEM)
# Analysis of performance of pile groups adjacent to deep excavation
# Influence of soil type on static response of cantilever sheet pile walls under surcharge loading: a numerical
study
#Behavior of anchored sheet pile wall.
# Energy sheet pile walls–Experimental and numerical investigation of innovative energy geostructures.
6. Following papers to be cited for developing section- need for future research before conclusion
# Reliability Analysis of Gravity Retaining Wall Using Hybrid ANFIS
#Reliability and prediction of embedment depth of sheet pile walls using hybrid ANN with optimization
techniques
# Probabilistic analysis of gravity retaining wall using ANFIS-based optimization techniques.
#Probabilistic design of retaining wall using machine learning methods
7. Grammar and spelling to be checked for complete manuscript

Reviewer 2 Report
The paper is well-organized and easy to understand. However some specific comments are as follows:
Introduction:
The introduction is well conducted. It addresses the issue and refers to relevant literature. However, please provide more examples (papers). What is the hypothesis of this paper?
Similarity theory and model test parameters
This chapter is well conducted, however Figs 3, 6 could be improved.
Results and Discussion:
The discussion should be completed. Please add more works because the purpose of the discussion is to review the study findings in light of the published literature.
Conclusions:
This chapter is well conducted.
Round 2
Reviewer 1 Report
Corrections have been made as per previous review.
Final manuscript can be submitted.
